# Investigation of the Rheological Properties and Storage Stability of Waste Polyethylene/Ethylene–Vinyl Acetate-Modified Asphalt with Crosslinking and a Silicone Coupling Agent

**DOI:** 10.3390/ma16093289

**Published:** 2023-04-22

**Authors:** Yuhao Ma, Tao Zhou, Hao Song, Hong Zhang

**Affiliations:** 1Department of Chemical Engineering, Northwest Minzu University, Lanzhou 730000, China; 2Key Laboratory of Environment-Friendly Composite Materials of the State Ethnic Affairs Commission, Lanzhou 730000, China; 3Gansu Provincial Biomass Function Composites Engineering Research Center, Lanzhou 730000, China; 4Key Laboratory for Utility of Environment-Friendly Composite Materials and Biomass in University of Gansu Province, Lanzhou 730000, China; 5Gansu Provincial General Station of Agricultural Ecology and Resource Conservation Technology Extension, Lanzhou 730000, China

**Keywords:** PE, EVA, DCP, KH-570, rheological analysis, storage stability

## Abstract

As the market for polyethylene consumption continues to expand, the amount of waste polyethylene is also increasing. Modifying asphalt with waste polyethylene (PE) is economical and environmentally friendly. The low-temperature performance and storage stability of PE-modified asphalt has long been an insurmountable problem. The high vinyl acetate (VA) content of ethylene–vinyl acetate (EVA) and PE blended into asphalt can improve the compatibility of PE and asphalt. It compensates for the high VA content of EVA brought about by the lack of high-temperature resistance to permanent deformation but is still not conducive to the stable storage of PE at high temperatures. The effect of furfural extraction oil, a crosslinking (DCP) agent, a silicone coupling agent (KH-570), and calcium carbonate (CaCO_3_) on the rheological properties and compatibility of PE/EVA-modified asphalt was investigated in this study. The conventional physical properties of PE/EVA-modified asphalt were tested after introducing furfural extraction oil, DCP, KH570, and CaCO_3_ to determine the correlations of these materials. In addition, frequency sweep, multiple stress creep and recovery (MSCR), and linear amplitude sweep (LAS) were utilized to characterize the rheological properties and fatigue behavior. The results reveal that the addition of suitable ratios of furfural extract oil, DCP, KH-570, and CaCO_3_ to PE/EVA-modified asphalt produces a remarkable improvement in the viscoelastic characteristics and viscosity compared with PE/EVA-modified asphalt. Furthermore, fluorescence microscopy (FM) was utilized to evaluate the modification mechanism, which shows that PE/EVA undergoes significant crosslinking in asphalt, forming a three-dimensional network structure that dissolves in the asphalt. The storage stability of the PE-modified bitumen was fully determined, and its high-temperature rheology was substantially improved.

## 1. Introduction

Asphalt is an organic colloidal mixture of lightweight components, such as various small hydrocarbon compounds and some unsaturated aromatic hydrocarbons, and is one of the oldest engineering materials. Refined asphalt, most of which was used in road construction, was first produced in the United States in the early 20th century. However, over the last few decades, continually increasing traffic volumes, heavier vehicle loads, unexpected weather extremes, and slight variations in ground conditions have posed major challenges to the consistency of pavements. Indeed, these challenges are mainly due to the inadequate mechanical properties of asphalt [1,2,3]. Considering that it is difficult for small-hydrocarbon compounds in asphalt to chemically react with other substances, and thus improve their mechanical properties, polymers usually modify asphalt to enhance its viscoelasticity. Numerous studies and methods have shown [1,4,5] that the modification of asphalt using polymers effectively solves problems such as rutting, cracking, and fatigue. Polyethylene waste can produce modified asphalt at a low cost, mitigate environmental pollution, enable the upgrade of energy-efficient construction technologies, and develop green construction projects [6,7,8,9,10]. In the spirit of environmental protection and lower costs, research on the use of waste polyethylene as modifiers to produce modified asphalt shows no sign of slowing down. PE is a kind of thermoplastic resin polymer that is commonly used for the production of films, tubes, wires, cables, and moldings. The residual pollution of used agricultural films has become one of the outstanding problems restricting the green development of agriculture in China. The capacity of agricultural waste films to be consumed must be improved due to the high impurity of agricultural waste film, low utilization value of recycled materials, and large-scale production. There is a need to continue to design and develop industrial products using agricultural waste film as a raw material [11]. PE-modified asphalt has increased rigidity and enhanced resistance to permanent deformation at high temperatures [12,13]. However, non-polar crystalline PE is less compatible with asphalt and is incorporated into the asphalt as a modifier. It has a poor low-temperature ductility and is prone to phase separation [14,15]. Many researchers have explored the storage stability problems of PE-modified asphalt, but these efforts remain in the ready-to-use stage of the application process. According to previous studies, EVA has been extensively studied in asphalt blended with PE to improve the phase separation of PE in asphalt [16,17].

EVA is a thermoplastic resin polymer modifier consisting of the copolymerization of ethylene and vinyl acetate, obtained by modifying the vinyl segment with the polar functional group vinyl acetate to increase the polarity of the vinyl segment and reduce its crystallinity, thus providing excellent flexibility and improving its compatibility with asphalt [18]. EVA has a similar molecular structure to PE and, according to the principle of similar solubility, is very compatible with PE. Adding EVA can effectively improve the storage stability and low-temperature ductility of PE-modified asphalt [16,17,19]. However, the addition of EVA alone is insufficient to completely solve the storage stability problem of PE-modified asphalt.

The optimal state of compatibility between polymer and asphalt occurs when the polymer is thoroughly dispersed in the asphalt in the most diminutive possible form, where it is instantaneously transformed from a discontinuous state to a continuous state, creating a rigid three-dimensional network structure. Good compatibility can considerably delay phase separation and thus improve storage stability, which significantly impacts the performance of the final modified asphalt [15,20,21]. To enhance the compatibility of PE/EVA composites with asphalt, attempts were made to add DCP and KH-570 to induce crosslinking reactions between PE/EVA composites via heating to form a three-dimensional network structure in the asphalt phase [22,23]. CaCO_3,_ which has the advantages of high strength and thermal stability and thus is often used to improve the mechanical properties of polyolefins, can further enhance the deficiencies in the mechanical properties of PE/EVA-modified asphalt [24,25]. The three-dimensional network structure formed by the crosslinking reaction of PE/EVA composites may encapsulate CaCO_3_, achieving a uniform dispersion of CaCO_3_ in the asphalt. As a result, the toughness, stiffness, and heat resistance of the PE/EVA-modified asphalt are improved. At the same time, the polarity of the PE/EVA increases, which significantly improves the dispersion of the PE/EVA in the base asphalt and further improves the compatibility of the PE/EVA with the asphalt. The addition of excess PE and EVA can disrupt the state of colloidal equilibrium in the base asphalt due to the spatial organization and arrangement of the SARA fractions present in the base asphalt themselves [26,27], so the choice of adding furfural extraction oil can make up for the loss of lighter components in the modified asphalt, allowing the colloid to return to equilibrium and avoiding the problem of an excess of PE/EVA that is not very compatible with the asphalt [28,29].

In a comprehensive consideration of the high- and low-temperature performance of PE/EVA-modified asphalt, this study utilized asphalt modified with 4% PE and 4% EVA to investigate the effect of furfural extract oil, DCP, KH-570, and CaCO_3_ on the rheological properties, fatigue resistance, and storage stability of PE/EVA-modified asphalt. The aim was to completely solve the high-temperature storage stability of PE-modified asphalt while further improving the high-temperature deformation resistance and fatigue resistance of PE/EVA-modified asphalt, providing a theoretical basis for a non-dissociation preparation scheme for PE/EVA-modified asphalt. The effects of furfural-extracted oil, DCP, KH-570, and CaCO_3_ on the conventional physical properties of PE/EVA-modified asphalt were determined using orthogonal experiments and grey correlation analysis, and the rheological properties and fatigue resistance of PE/EVA-modified asphalt were investigated using dynamic shear rheology (DSR) tests. DSR tests include a frequency sweep at 25 °C, multiple stress creep and recovery (MSCR) at 64 °C, and a linear amplitude sweep (LAS). Furthermore, the phase distribution between the polymer modifier and asphalt was investigated using fluorescence microscopy (FM).

## 2. Experimental and *Methods*

### 2.1. Raw Materials

Base asphalts (denominated as “SK-90”) with penetration grade 90 were used in this research. The properties of base asphalt and copolymers are listed in Table 1 and Table 2, respectively.

γ-Methacryloxypropyltrimethoxysilane (KH-570), obtained from Aladdin Corp. (Beijing, China), was chosen for its abilities to chemically react with polyolefins initiated by peroxide and graft onto a polymer surface to improve the bonding strength and low-temperature resistance of the polymer. The dicumyl peroxide (DCP) used in this study was also supplied by Aladdin Corp. (Beijing, China). It was chosen because it can cross-link polyolefins under certain conditions, forming a three-dimensional network structure that can significantly improve the heat resistance, cohesive strength, adhesive strength, solvent resistance, and other properties of polyolefins. The furfural extract oil was offered by Shandong Longsheng da New Material Technology Co., Ltd. (Zibo, Shandong, China). CaCO_3_ was purchased from Linyi Quanlin Chemical Co., Ltd. (Linyi, Shandong, China).

### 2.2. Preparation of Furfural Extract Oil/KH-570/DCP/CaCO_3_/PE/EVA-Modified Asphalt Samples

Firstly, the base asphalt was heated to 160 ± 5 °C in an oil bath to soften the asphalt to a fluid state. The weighed PE, EVA, and furfural extraction oil were separately poured into the base asphalt, preheated, and sheared at 2000 r/min for 5 min with a high-speed shear to produce PE, EVA, and furfural extraction oil, which were initially melted and dispersed to prevent accumulation. After that, the temperature and shear rate were increased to 180–190 °C and 5000 r/min, respectively, and the mixture was sheared again for 60 min. Finally, the modified asphalt obtained from the high-speed shear was cooled down to 160 ± 5 °C. DCP, KH-570 and CaCO_3_ were added one-by-one, and the temperature was maintained at 175 °C in a temperature-controlled electric heating jacket. The developed mixture was stirred with an electric mixer for one hour to obtain PE/EVA-modified asphalt. The process of preparing specimens is displayed in Figure 1.

### 2.3. Test Methods

Firstly, the orthogonal experiment determined nine sets of experimental matching parameters. Then, the specimens were prepared based on the nine sets of experimental matching parameters to evaluate the effects of furfural extract oil, DCP, KH-570, and CaCO_3_ on the conventional physical and rheological properties of PE/EVA-modified asphalt. Table 3 represents the traditional physical tests for asphalt modified by 4% PE and 4% EVA. Figure 2 displays a flow diagram of the experiment.

#### 2.3.1. Orthogonal Design of Experiments

Firstly, the factors for the experiment were determined, and appropriate levels were selected. The identification of factors and the choice of levels depended on the empirical summary of the investigation. The orthogonal experiments were designed so that, although the parameter ratios did not result in an overall optimal setting, the simplified arrays were still statistically significant, fast, and accurate. The effect of different ratios of furfural extract oil, DCP, KH-570, and CaCO_3_ on the conventional physical properties of PE/EVA-modified asphalt was investigated using grey correlation analysis. An L_9_(_34_) orthogonal table was used to arrange the four factors, with three levels for each part. Factor-level and orthogonal arrays are shown in Table 4 and Table 5, respectively.

#### 2.3.2. Conventional Physical Properties Tests

The physical properties of modified asphalt, including softening point, penetration at 25 °C, and flexibility at 5 °C, were evaluated in ASTM D36, D4402, and D113, respectively.

#### 2.3.3. Dynamic Shear Rheometer Tests

Using the dynamic shear modulus and phase angle, as measured by dynamic shear rheometer (DSR) tests, is currently the most widely used way to evaluate the rheological properties of asphalt. This study used three rheological tests—multiple stress creep and recovery (MSCR), the frequency sweep test after short-term aging, and the linear amplitude sweep (LAS) after short-term aging—to assess the rheological properties and fatigue life of the modified asphalt. The parallel plate molds with a diameter of 8 mm were used when the test temperature was 30 °C and below, and 25 mm plate molds were used when the test temperature was above 30 °C. Two creep stress levels of 0.1 kPa and 3.2 kPa were chosen for the MSCR. Ten cycles of each stress level were conducted, each consisting of a 1 s creep phase and a 9 s unloading recovery phase, for a total test time of 200 s, to obtain the average creep-recovery-rate R value and the average irrecoverable creep flexibility J_nr_ value. The lower the J_nr_ value, the better the resistance of the modified asphalt to permanent deformation. The frequency sweep (0.1–100 rad/s) test maintains a 1% shear strain at a temperature of 25 °C. The resulting dynamic shear modulus (|G*|) responds to the viscoelasticity of the asphalt in the frequency domain (ω) at different temperatures. |G*| is the ratio of maximum stress to maximum strain, usually used to characterize asphalt resistance to deformation, and δ reveals the viscoelastic behavior of asphalt. The LAS comprises an initial frequency sweep test followed by the amplitude sweep test. The initial frequency sweep test with a strain amplitude of 0.1% was measured for 25 °C from 0.1 Hz to 30 Hz, which is used to obtain the undamaged material characteristics and the α parameter. The amplitude sweep test, at 10 Hz and 25 °C, is a strain-control loading mode. The dynamic-loading amplitude sinusoidal functions has a linear growth from 0.1 Hz to 30 Hz over 3100 cycles of loading. The test time was 5 min. The fatigue life of different kinds of modified asphalt was calculated based on the continuous viscoelastic damage (VECD) mechanics model (Equation (1)) [30]. Higher $N_{f}$ means higher fatigue life.

Fluorescence microscopy (FM) images can visualize the distribution of the polymer phase with the asphalt phase. A drop of hot asphalt sample was sandwiched between a slide and coverslip, placed on a fluorescent microscope stage, and the selection was observed at a magnification of 400,000 to monitor the dispersion of the polymers in the asphalt.

The high-temperature storage test was designed to assess the aggregation of polymers under high-temperature storage in asphalt to determine the storage stability of modified asphalt. Next, 50 g of samples were poured into a standardized aluminum tube, then vertically placed in an oven at 163 ± 1 °C for 48 h, after which the line was removed from the range and placed in a refrigerator for four h. The aluminum tube was divided into three equal parts, and the softening point test was carried out on the upper and lower parts of the sample. The difference in softening point was measured to assess the storage stability of modified asphalt. The smaller the difference in softening point, the better its high-temperature storage stability.
(1)$$N_{f}=\gamma_{max}{}^{-B}$$
where

A and B: VECD model parameters related to material properties;$N_{f}$: Fatigue failure life;$\gamma_{max}$: Maximum shear strain for the given pavement structure.

## 3. Results and Discussion

### 3.1. Grey Relation Analysis

The grey relation analysis results of the orthogonal experiment are listed in Table 5, Table 6, Table 7 and Table 8. The test parameters were used as the parent sequence, and the furfural extracts oil, DCP, KH-570, and CaCO_3_ as the subseries. The subsequence and the parent sequence were first dimensionless, utilizing averaging. The absolute differences between the corresponding elements of subsequences and the parent sequence were calculated one by one to determine the correlation coefficients, where ρ was taken as 0.5 is the discrimination factor. Finally, the average values of the correlation coefficients for the corresponding elements of subsequences and the parent sequence were obtained to reflect the correlation relationship, which is called the correlation sequence, denoted as r_0i_ (i = 1, 2, …, n). The results of the analysis were based on the correlation sequence.

Table 6, Table 7, Table 8 and Table 9 list the effect of the ratio of different parameters on the softening point, flexibility at 5 °C, penetration at 25 °C, and softening point difference of asphalt binders. As shown in Table 6, Table 7 and Table 8, according to the r_0i_ values, the factors that affect the softening point, flexibility, and penetration were furfural extract oil, KH-570, CaCO_3,_ and DCP, in that order. As shown in Table 9, furfural extraction oil was the most significant factor affecting the softening point difference, with the r_0i_ value for DCP (0.6674) exceeding KH-570 (0.6637) and CaCO_3_ (0.6637), which indicates that DCP had a more substantial influence on the storage stability than KH-570 and CaCO_3_. The degree of leverage of each of the four fillers on the different properties of the samples was determined from the results of a grey correlation analysis, which showed the effects of different ratios of the four fillers on the conventional physical properties of the PE/EVA-modified asphalt.

As shown in Table 6, Table 7, Table 8 and Table 9, in the nine sets of experiments, the samples with a higher ratio of DCP to KH-570 had better overall conventional physical properties. The pieces with more furfural-extracted oil tended to decrease in softening point and softening point difference and increase in elasticity and needle penetration. While CaCO_3_ was raised, there was a tendency for the softening end of the samples to rise and the elasticity and needle penetration to fall [31]. Nonetheless, the effect on the difference in softening point was insignificant and may have depended on the degree of crosslinking of PE and EVA. It is worth noting that when 1.4% of furfural-extraction oil, 0.03% DCP, 0.01% KH-570, and 0.05% CaCO_3_ were added, the difference in softening point was reduced by 12.3 °C, the ductility increased by 6.3 cm, the softening end increased by 2.2 °C, and the penetration decreased by 4.6 (0.1 mm) compared to when no filler was added.

The results of orthogonal experiments and grey correlation analysis show that furfural extracted oil is the most crucial factor affecting the conventional physical properties of PE/EVA-modified asphalt. Furfural extraction oil is composed of complex hydrocarbons with high levels of cycloalkanes and aromatic hydrocarbons, which are chemically similar to the base asphalt’s SARA fraction. Hence, it can supplement the lighter components of asphalt. The similar polarity of EVA and base asphalt allows EVA to interact and migrate with the asphalt phase, prompting EVA to dissolve in the lighter components of the asphalt. According to the principle of similar solubility, the identical molecular structures of EVA and PE cause some of the PE to dissolve in the asphalt. Excess polymer competed with the colloid in the asphalt for the lighter components, which broke up the arrangement of the SARA fraction in the asphalt itself, not fully exploiting the properties of the polymer. Adding the furfural extraction oil may bring the whole mixture back to an alternative colloidal equilibrium, reducing the degree of polymer agglomeration and softening the hard asphalt, thus enhancing elasticity and improving flexibility. Aromatic fractions are very sensitive to temperature [29,32] and prone to aging, making the softening point lower and the penetration higher.

The weak peroxide bond (O-O) in DCP was cleaved by heating to generate free radicals, which attacked the hydrogen ions on the carbon chain of PE and EVA, causing PE and EVA to become macromolecular free radicals due to the loss of hydrogen ions. These ions combined with each other to cause PE and EVA to crosslink and form a three-dimensional network structure [23]. In the same way, DCP can graft KH-570 onto ethylene chains to produce polymers containing ethyl silyl ester groups. The addition of KH-570 at a level less than that of DCP may slow down the rate of crosslinking of the polymer caused by DCP, preventing the bursting of macromolecular radicals caused by prolonged development at high temperatures, while KH-570 grafted on ethylene segments may also increase the polarity of the polymers and the strength of the modified asphalt. When more KH-570 was added than DCP, a weak crosslinking of the polymers may have occurred, significantly affecting the viscoelasticity and storage stability of the modified asphalt.

Suitable ratios of DCP and KH-570 may allow the samples to form a perfect three-dimensional network structure. The polymer was dissolved in the asphalt to create a solution of polymers, and the mechanochemical method was used to induce crosslinking of the polymer on the surface of the CaCO_3_ to achieve encapsulation of the CaCO_3_, increasing the compatibility between the CaCO_3_ and the polymers and improving the viscoelasticity and storage stability of the modified asphalt.

### 3.2. MSCR Creep Compliance

In the MSCR test, the effect of damage to the material at high temperatures can be determined [33]. The average of 10 cycles at a temperature of 64 °C was used as the creep recovery rate (R0.1 and R3.2) and the irrecoverable creep flexibility (J_nr0_._1_ and J_nr3_._2_) at each stress level, where the lower the J_nr_ value, the better the resistance of the modified asphalt to permanent deformation.

Corresponding J_nr_ and R values of different kinds of modified asphalt at the stress level of 0.1 kPa and 3.2 kPa and temperature of 64 °C are shown in Figure 2 and Figure 3. At a temperature of 64 °C, elastic recovery significantly decreased when the pressure was increased from 0.1 kPa to 3.2 kPa. Asphalt exhibits more viscous properties at high temperatures and pressures. As expected, the R0.1 and R3.2 of sample 8 increased by 34.3% and 60.4%, respectively, and the J_nr0_._1_ and J_nr3_._2_ decreased by 21.3% and 8.78%, compared to the unfilled sample, indicating that sample 8 has not only better elasticity but also better resistance to permanent deformation at high temperatures and pressures. Samples 1, 4, and 7, which were all supplemented with 0.01% DPC, had the worst R and Jnr values of all samples, most likely because the DCP was too small to trigger the grafting of KH-570 with the polymer and the crosslinking of the PE and EVA, prompting parts of the KH-570 and CaCO_3_ to become free in the asphalt, which had more significant side effects on viscoelasticity at high temperatures. When 0.05% DCP was added, there was a tendency for the elasticity of the samples to increase with little change in its resistance to permanent deformation, most likely due to the large area of crosslinking of the polymers caused by the DCP.

### 3.3. Linear Amplitude Sweep (LAS) Test Results

LAS is a test method for the fatigue performance of asphalt under dynamic loading. In the LAS, the fatigue life of asphalt is analyzed and predicted at strain levels of 2.5% and 5% based on the stress and strain mechanical responses of the asphalt obtained at moderate temperatures, with the damage criterion defined as a 35% reduction in the initial modulus of the asphalt.

The load–displacement curves for different kinds of modified asphalt are shown in Figure 4. It is evident that the shear stress of all binders increased during the initial loading phase and then decreased with the increase in shear strain. The slope of sample 7 is the lowest of all models, most likely because when the ratio of DCP to KH-570 was too small, the DCP was insufficient to trigger the involvement of KH-570 in the reaction, which disturbed the internal structure of the asphalt.

The fatigue life of different kinds of asphalt was obtained based on the continuous viscoelastic damage (VECD) theory, and the corresponding results are shown in Figure 5. The fatigue life of asphalt binder substantially decreased when the applied strain increased from 2.5% to 5%. This can be attributed to the magnitude of the stress at a strain of 2.5%, which did not fully activate the complex polymer structure. As the strain level increased in the LAS, an improved correlation could be observed that may be related to an accelerated shortening of fatigue life. This is because higher strain levels activate the complex structure of the polymer, leading to significant changes in the base asphalt. Elastomeric materials are generally subjected to less force at failure than plastic bodies, which are more complicated. It is observed in the graph that the smaller the ratio of DCP to KH-570, the higher the fatigue life, most likely because KH-570 was grafted onto the carbon chain of the polymer with DCP as initiator, which increased the hardness of the polymer, reduced its elasticity, and provided better wear resistance at strain force levels. The samples had a lower fatigue life at 0.05% DCP, most likely because the polyethylene was crosslinked to give the modified asphalt better elasticity.

### 3.4. Complex Shear Modulus and Phase Angle

A frequency sweep test was conducted to determine the linear viscoelastic mechanical properties of modified asphalt under specific temperature and frequency conditions. The frequent sweep results of the modified asphalt are presented in Figure 6.

The G* decreased and increased with the increase in frequency, indicating an increase in vehicle speed, and asphalt has more rigid and elastic properties. In Figure 6a–c, it is evident that the G* and δ of sample 9 are smaller than the base asphalt. Once again, a viscoelastic failure prompted by a small DCP to KH-570 ratio is demonstrated. In addition, samples 2, 4, and 8 have relatively large G* and small δ, indicating a reduction in shear deformation resistance above 0.03% for DCP or KH-570 and a reduction in the degree of transition from viscous body to elastomer. The overall higher δ observed in Figure 6b,c compared to Figure 6a suggests that the increase in furfural extracted oil was able to improve the elasticity of the modified asphalt.

### 3.5. Fluorescence Microscopy Image Analysis

The microstructure of one modified asphalt (4% PE + 4% EVA) and another modified asphalt (4% PE + 4% EVA + 1.4% furfural extracted oil + 0.03% DCP + 0.01% KH-570 + 0.05% CaCO_3_) were observed using fluorescence microscopy (FM). FM images are shown in Figure 7a,b, where black and yellow represent the asphalt and polymer phases, respectively.

Figure 7a shows that PE and EVA were present as large particles in the asphalt phase and inevitably caused phase separation of the polymer from the asphalt under high-temperature storage due to the difference in density and polarity. In Figure 7b, it is clear that the polymer forms a filamentary network structure in the asphalt phase, effectively preventing polymer aggregation. The FM images further explain the co-mingling mechanism after the addition of furfural extract oil, DCP, KH-570, and CaCO_3_, demonstrating the critical reasons for the improved viscoelasticity and storage stability of the asphalt.

## 4. Conclusions

In this study, the effects of different ratios of furfural-extraction oil, DCP, KH570, and CaCO_3_ on the conventional physical properties of PE/EVA-modified asphalt, as well as the rheological properties at medium and high temperatures, were explored. In addition, fluorescence microscopy was used to assess morphology and microstructure. A research method for the preparation of PE/EVA-modified asphalt without segregation was obtained. The following conclusions can be drawn.

(1) Based on orthogonal experiments and grey correlation analysis, the factors that affected the softening point, flexibility, and penetration were furfural extract oil, CaCO_3_, DCP, and KH-570, in that order. However, KH-570 had a greater effect on the storage stability of samples than DCP. Furfural-extracted oil improved the ductility and needle penetration of PE/EVA-modified bitumen and reduced the softening point and softening point difference. The addition of CaCO_3_ increased the softening point and decreased the ductility and needle penetration, but the effect on the difference in softening point was not significant, depending on the degree of cross-linking of PE and EVA. It is worth noting that, when 1.4% furfural extraction oil, 0.03%DCP, 0.01%KH-570, and 0.05%CaCO_3_ were added, the difference in softening point was reduced by 12.3 °C, the ductility increased by 6.3 cm, the softening end increased by 2.2 °C, and the penetration decreased by 4.6 (0.1 mm) compared to when no filler was added. 

(2) MSCR tests showed that the addition of DCP significantly improved the elastic recovery of PE/EVA-modified asphalt under repeated negative traffic loading at high temperatures. The addition of an appropriate amount of KH-570 further improved the viscoelasticity of the modified asphalt at high temperatures, but too large or too small a proportion of DCP to KH-570 had a negative impact on the viscoelasticity of the modified asphalt. More importantly, higher viscoelasticity is beneficial for enhancing the resistance of the asphalt mix to permanent deformation at high temperatures.

(3) The sweep tests showed that the shear deformation resistance of PE/EVA-modified asphalt was improved and the elasticity was enhanced at 0.01–0.03% of DCP and KH-570 in the case of vehicle speed increase at a medium temperature. The increase in furfural-extracted oil slowed down the degree of transition from viscosity to elasticity of the modified asphalt. LAS tests showed that KH-570 significantly improved the wear resistance and enhanced the fatigue life of the PE/EVA-modified asphalt. dCP significantly improved the elasticity of the PE/EVA-modified asphalt, leading to a reduction in fatigue life.

(4) The phase distribution of the polymers in the asphalt was examined using fluorescence microscopy. Under the condition of DCP as initiator, the addition of KH-570 can induce PE and EVA to cross-link, and its fine filamentous mesh structure can wrap CaCO_3_, promoting the uniform dispersal of CaCO_3_ in the asphalt phase and significantly improving the storage stability of PE/EVA-modified asphalt at high temperatures. 

## Figures and Tables

**Figure 1 materials-16-03289-f001:**
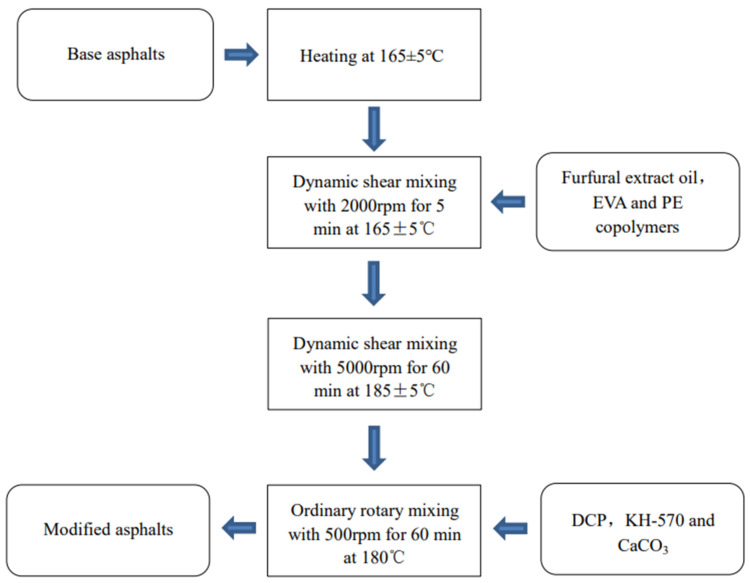
Preparation process of specimens in the laboratory.

**Figure 2 materials-16-03289-f002:**
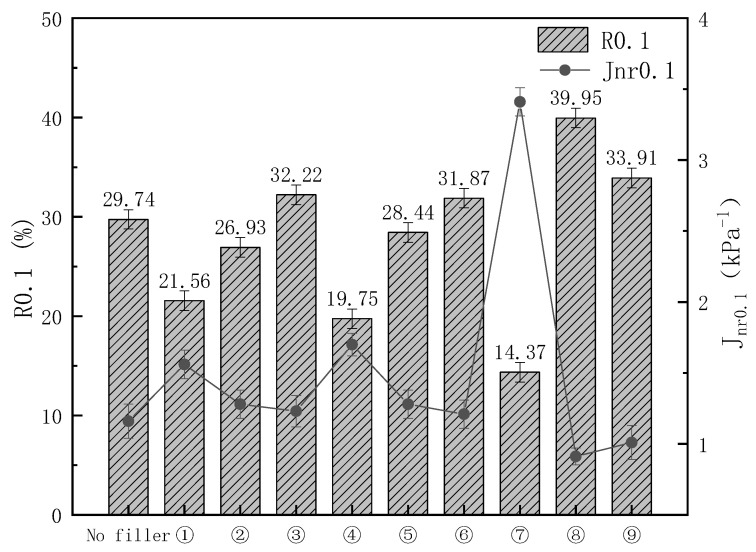
Nonrecoverable creep compliance and percent recovery at the stress level of 0.1 kPa.

**Figure 3 materials-16-03289-f003:**
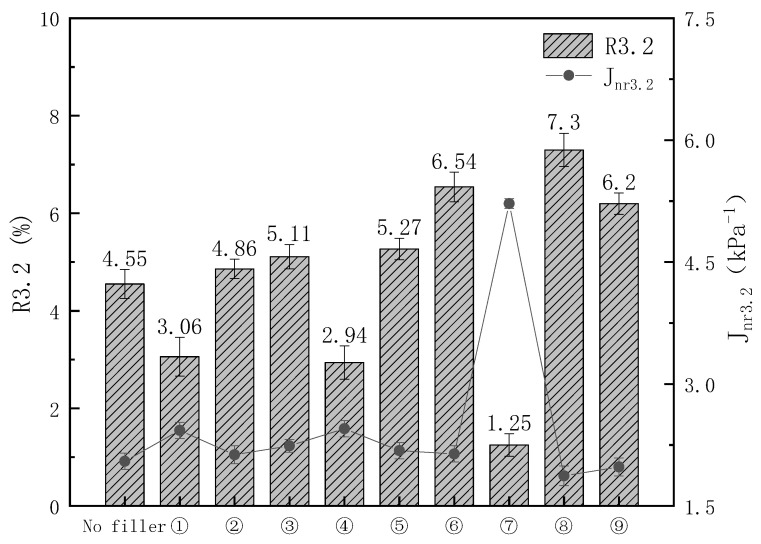
Nonrecoverable creep compliance and percent recovery at the stress level of 3.2 kPa.

**Figure 4 materials-16-03289-f004:**
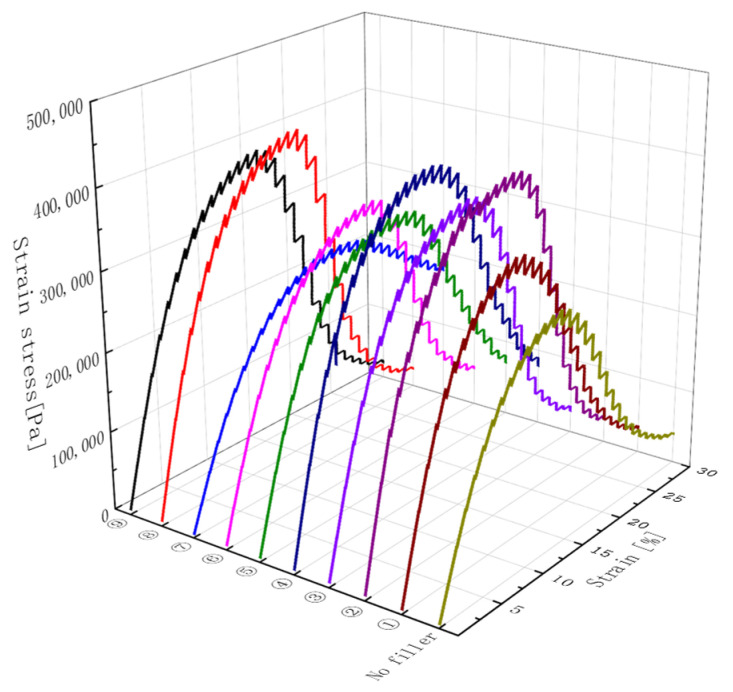
The relationship between shear stress and shear strain during LAS tests.

**Figure 5 materials-16-03289-f005:**
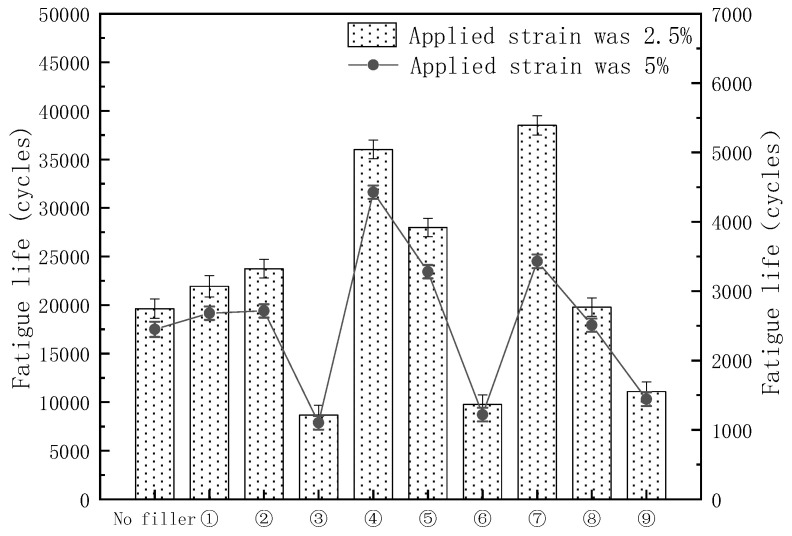
Fatigue life of different kinds of modified asphalt under the conditions of two applied strains.

**Figure 6 materials-16-03289-f006:**
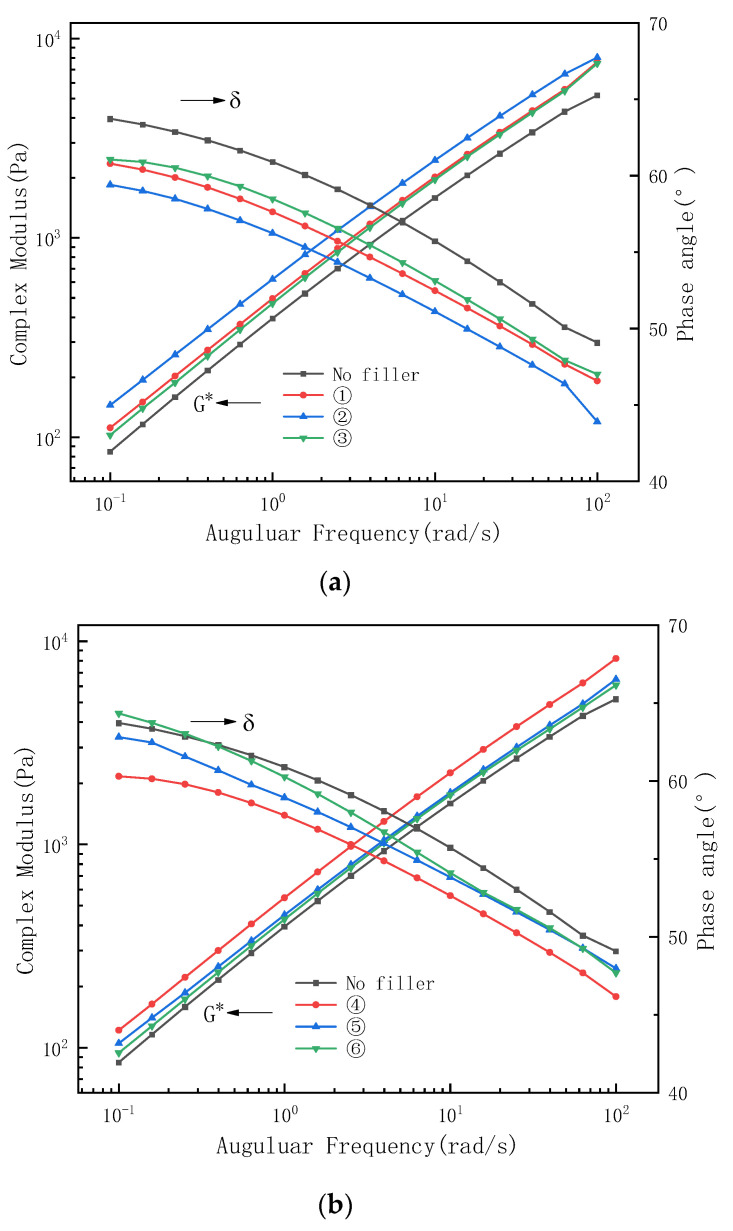
G* and δ results of temperature sweep test. (**a**) Test Number ①–③; (**b**) Test Number ④–⑥; (**c**) Test Number ⑦–⑨.

**Figure 7 materials-16-03289-f007:**
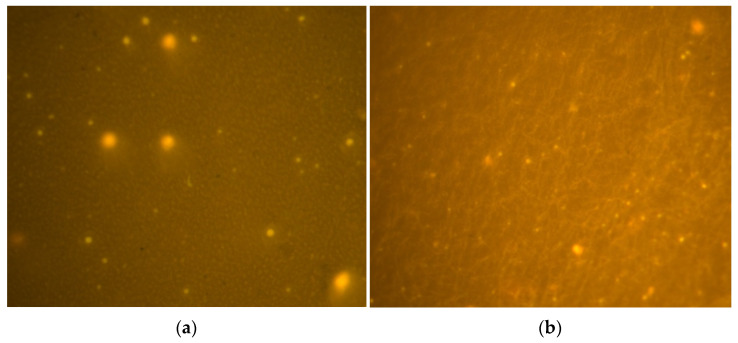
Fluorescence microscope images of modified asphalts. (**a**) modified asphalt (4% PE + 4% EVA); (**b**) modified asphalts (4% PE + 4% EVA + 1.4% furfural extracted oil + 0.03% DCP + 0.01% KH-570 + 0.05% CaCO_3_)

**Table 1 materials-16-03289-t001:** The properties of the base asphalt.

Items	Test Temperature (°C)	Value
Penetration (0.1 mm)	25	83
Softening point (°C)	-	47
Ductility (cm)	15	180
Flash point (°C)	-	304
Solubility (%)	-	99.8
Density(g/cm^3^)	15	1.030

**Table 2 materials-16-03289-t002:** The properties of EVA and PE copolymers.

Property	Value	
	PE	EVA
Density at 25 °C (g/cm^3^)	0.91	0.948
Melting point (°C)	122	75
Melt Index (dg/min)	1.8	65

**Table 3 materials-16-03289-t003:** The properties of 4% PE + 4% EVA-modified asphalt.

Items	Test Temperature (°C)	Value
Penetration (0.1 mm)	25	61
Softening point (°C)	-	54.6
Ductility (cm)	5	7.3
Softening point difference (°C)	-	12.4

**Table 4 materials-16-03289-t004:** The factors and levels in the orthogonal experiment.

	Factors			
Levels	A	B	C	D
	Furfural Extract Oil	DCP	KH-570	CaCO_3_
	(g)	(g)	(g)	(g)
1	5.0	0.05	0.05	0.05
2	6.0	0.15	0.15	0.15
3	7.0	0.25	0.25	0.25

**Table 5 materials-16-03289-t005:** Test program.

	Factors				
Test Number	A	B	C	D	Test Program
①	1	1	1	1	A_1_B_1_C_1_D_1_
②	1	2	2	2	A_1_B_2_C_2_D_2_
③	1	3	3	3	A_1_B_3_C_3_D_3_
④	2	1	2	3	A_2_B_1_C_2_D_3_
⑤	2	2	3	1	A_2_B_2_C_3_D_1_
⑥	2	3	1	2	A_2_B_3_C_1_D_2_
⑦	3	1	3	2	A_3_B_1_C_3_D_2_
⑧	3	2	1	3	A_3_B_2_C_1_D_3_
⑨	3	3	2	1	A_3_B_3_C_2_D_1_

**Table 6 materials-16-03289-t006:** The influence of the ratio of different parameters on the softening point of modified asphalt.

	Factors				Softening Point
Test Number	A	B	C	D	(°C)
①	1	1	1	1	56.1
②	1	2	2	2	57.5
③	1	3	3	3	59.8
④	2	1	2	3	58
⑤	2	2	3	1	57.2
⑥	2	3	1	2	56.3
⑦	3	1	3	2	58.1
⑧	3	2	1	3	56.8
⑨	3	3	2	1	57.9
r_0i_	0.772	0.556	0.560	0.553	
Order of association: r_01_ > r_03_ > r_02_ > r_04_					

**Table 7 materials-16-03289-t007:** The influence of the ratio of different parameters on the flexibility of modified asphalt.

	Factors				Ductility (5 °C)
Test Number	A	B	C	D	[32]
①	1	1	1	1	6.4
②	1	2	2	2	7.6
③	1	3	3	3	7.8
④	2	1	2	3	6.7
⑤	2	2	3	1	7.4
⑥	2	3	1	2	8.4
⑦	3	1	3	2	8.2
⑧	3	2	1	3	13.6
⑨	3	3	2	1	11.5
r_0i_	0.850	0.629	0.579	0.693	
Order of association: r_01_ > r_04_ > r_02_ > r_03_					

**Table 8 materials-16-03289-t008:** The influence of the ratio of different parameters on the penetration of modified asphalt.

	Factors				Penetration (25 °C)
Test Number	A	B	C	D	(0.1 mm)
①	1	1	1	1	54.4
②	1	2	2	2	53.2
③	1	3	3	3	50.4
④	2	1	2	3	53.4
⑤	2	2	3	1	56.5
⑥	2	3	1	2	58.2
⑦	3	1	3	2	56.3
⑧	3	2	1	3	56.8
⑨	3	3	2	1	55.2
r_0i_	0.804	0.555	0.559	0.546	
Order of association: r_01_ > r_03_ > r_02_ > r_04_					

**Table 9 materials-16-03289-t009:** The influence of the ratio of different parameters on the softening point difference of modified asphalt.

	Factors				Softening Point Difference
Test Number	A	B	C	D	(°C)
①	1	1	1	1	11.2
②	1	2	2	2	8.3
③	1	3	3	3	6.1
④	2	1	2	3	6.6
⑤	2	2	3	1	8.6
⑥	2	3	1	2	5.2
⑦	3	1	3	2	6.4
⑧	3	2	1	3	0.1
⑨	3	3	2	1	1.5
r_0i_	0.727	0.556	0.684	0.677	
Order of association: r_01_ > r_03_ > r_04_ > r_02_					

## Data Availability

All data required for reproducibility are provided within the paper.

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
