# Peer review of "Investigation of the Rheological Properties and Storage Stability of Waste Polyethylene/Ethylene–Vinyl Acetate-Modified Asphalt with Crosslinking and a Silicone Coupling Agent"

_materials, 2023, doi:10.3390/ma16093289_

Round 1
Reviewer 1 Report
This paper evaluated the rheological properties, fatigue resistance, and storage stability performance of PE/EVA binders containing extract oil, DCP, KH-570, and CaCO3.
For these two-asphalt modifiers (PE/EVA), PE/EVA could increase the stiffness or elasticity of asphalt binder, while extract oil could decrease the stiffness of asphalt binder, so their benefits to modify asphalt binder might be compromised when they are jointly used. The authors do not justify what are the problems of using PE and EVA individually. The use of PE or EVA only might have better modification for binder and mixture, so that there might be no need to use the hybrid PE and EVA together. Authors need to explain the reason or conduct additional experiment to use PE or EVA only and show the problems.
Authors must be able to justify the novelty of their research in a separate section like 'goals and objectives'. There you should also describe the research limitations and contribution to the body of the literature.
knowledge gap should be clarified clearer.
in introdcution, line 75 , it was stated "However, the addition of EVA alone is insufficient to solve the storage stability problem of PE-modified asphalt completely." pls justify this sentence by referring to proper studies.
this study used the content of 4% for both PE and EVA. why this content was used?
- authors should completely write out the meaning of the abbreviation or acronym when you first use it, for example " LLDPE in line 66, DCP and KH-570 in line 84, CaCO3 in line86, also for the title, it is preferred to not use abbreviations in title.
- in materials, pls clarify more what are DCP and KH-570? what are used for? what are their properties ?
- in section of Preparation of modified asphalt samples, pls clarify more how the PE and EVA were mixed together ? also, DCP, KH-570 and CaCO3, how they were added to binder, were they added separately or together?
- pls change "one h" in line 132 to " one hour"
- in line 140. it is stated "Fig. 2 displays the flow diagram of the experiment program." pls check this fig as it looks it is missing in the text
- Table 4, are these content in gram? 0.01 g is too little value. also how the values were obtained?
- in section "2.3.3. Dynamic shear rheometer tests" authors are described DSR test but also other tests (FM . LAS, MSCR, and high-temperature storage) were stated under the same DSR test. pls use proper standard to refer to these tests.
- authors used Fig. and Figure interchangeably, it is advised to use either Fig. or Figure.
- authors are suggested to strengthen the discussion of the results as the effect of each constitute ( extract oil, DCP, KH-570, and CaCO3) were discussed
- pls revise the format of the references.
- Line 283, pls revise the order of figures number (Figures 2 and 4)
Author Response
Dear Ms. Irina Mariana and dear reviewer
Re: Manuscript ID: materials-2331236 and Title: Investigation of the rheological properties and storage stability of waste polyethylene/ethylene–vinyl acetate-modified asphalt with Crosslinking and Silicone Coupling Agent
Thank you for your letter and the reviewers’ comments concerning our manuscript entitled “materials-2331236” (ID). Those comments are valuable and very helpful. We have read through comments carefully and have made corrections. Based on the instructions provided in your letter, we uploaded the file of the revised manuscript. Revisions in the text are shown using yellow highlight for additions, and strikethrough font for deletions. We had sought the MDPI appointed company to do the English grammar changes and indicate in purple. The responses to the reviewer's comments are presented following:
Question 1:The authors do not justify what are the problems of using PE and EVA individually. The use of PE or EVA only might have better modification for binder and mixture, so that there might be no need to use the hybrid PE and EVA together. Authors need to explain the reason or conduct additional experiment to use PE or EVA only and show the problems.
Answer 1:Many researchers have used PE and EVA modified bitumen separately. For example, Liang Ming[1] supplied four types of polyethylene (HDPE, MDPE, LDPE, LLDPE) in pellet form and used fluorescence microscopy (FM) to observe the morphology of four different PE-modified bitumen at storage temperature, and evaluated their storage stability by test tube tests and rheological characterisation, as PEs are non-polar and highly crystalline, they are almost completely with bitumen They are almost completely immiscible with bitumen due to the non-polar and high crystallinity of PE.
Liang Ming[2] used dynamic shear rheometry (DSR), bending beam rheometry (BBR), storage stability tests and optical microscopy analysis by varying the vinyl acetate content in EVA. The results showed that EVA with low VA content had little effect on the improvement of viscoelasticity and the large dispersed particles led to poor storage stability. On the other hand, as the VA content of EVA increases, the rigidity decreases. The crystalline domains become smaller and are also easily dispersed in smaller sizes. In addition, more asphaltic species migrate into the intermolecular spaces of EVA, resulting in a well-swollen polymer-rich phase and a more interwoven physical network.
In the field of composite modified bitumen, Yan Kezhen[3] demonstrated that EVA with low vinyl acetate content not only improves thermal storage stability and resistance to deformation at high temperatures, but also weakens the effect of LDPE on the low temperature performance of bitumen.
Therefore, blending EVA and PE with high VA content into asphalt improves the compatibility of PE with asphalt and also compensates for the lack of high temperature resistance to permanent deformation brought about by EVA with high VA content. The two polymers complement each other.
We have included this sentence in the corresponding section of the introduction. “The high vinyl acetate (VA) content of ethylene–vinyl acetate (EVA) and PE blended into asphalt can improve the compatibility of PE and asphalt. It compensates for the high VA content of EVA brought about by the lack of high-temperature resistance to permanent deformation”.
Question 2: Authors must be able to justify the novelty of their research in a separate section like 'goals and objectives'. There you should also describe the research limitations and contribution to the body of the literature.
Answer 2: We had added “The aim was to completely solve the high-temperature storage stability of PE-modified asphalt, while further improving the high-temperature deformation resistance and fatigue resistance of PE/EVA-modified asphalt, providing a theoretical basis for a non-dissociation preparation scheme for PE/EVA-modified asphalt.” to the introduction and “A research method for the preparation of PE/EVA-modified asphalt without segregation was obtained” to the Conclusions.
Question 3:In introdcution, line 75, it was stated "However, the addition of EVA alone is insufficient to solve the storage stability problem of PE-modified asphalt completely." pls justify this sentence by referring to proper studies.
Answer 3:Many studies[4,5] have shown that the problem of high temperature storage stability of PE modified bitumen can be improved by EVA, but it is not a complete solution.
Question 4:This study used the content of 4% for both PE and EVA. why this content was used?
Answer 4:In the previous stage, 2% and 4% recycled PE and 2%, 4%, 6% and 8% EVA were used in different ratios to analyse the modification effect of PE/EVA modifiers on asphalt by comparison under the needle penetration, ductility and softening point tests. For the comprehensive consideration of the high and low temperature performance of PE/EVA modified asphalt, 4% PE with 4% EVA was selected for the preparation of modified asphalt. We had added “In a comprehensive consideration of the high- and low-temperature performance of PE/EVA-modified asphalt, this study utilized asphalt modified with 4% PE and 4% EVA” to the introduction.
Question 5:Authors should completely write out the meaning of the abbreviation or acronym when you first use it, for example " LLDPE in line 66, DCP and KH-570 in line 84, CaCO3 in line86, also for the title, it is preferred to not use abbreviations in title.
Answer 5:We have modified the title and the meaning of the abbreviation or acronym in full for first use.
Question 6: In materials, pls clarify more what are DCP and KH-570? what are used for? what are their properties ?
Answer 6:We had made changes in the material section.
Question 7:In section of Preparation of modified asphalt samples, pls clarify more how the PE and EVA were mixed together? also, DCP, KH-570 and CaCO3, how they were added to binder, were they added separately or together?
Answer 7:We have made changes in the sample preparation section.
Question 8:Pls change "one h" in line 132 to " one hour".
Answer 8:We have made changes where appropriate.
Question 9:In line 140. it is stated "Fig. 2 displays the flow diagram of the experiment program." pls check this fig as it looks it is missing in the text.
Answer 9:We have added the missing parts of Fig. 2.
Question 10:Table 4, are these content in gram? 0.01 g is too little value. also how the values were obtained?
Answer 10:We have made changes to Table 4. Our level selection was determined by means of a quantitative analysis.
Question 11:In section "2.3.3. Dynamic shear rheometer tests" authors are described DSR test but also other tests (FM, LAS, MSCR, and high-temperature storage) were stated under the same DSR test. pls use proper standard to refer to these tests.
Answer 11:In section "2.3.3. Dynamic shear rheometer tests" we had used appropriate criteria to reference these tests.
Question 12:Authors used Fig. and Figure interchangeably, it is advised to use either Fig. or Figure.
Answer 12:We have changed the Figure to Fig.
Question 13: Authors are suggested to strengthen the discussion of the results as the effect of each constitute ( extract oil, DCP, KH-570, and CaCO3) were discussed.
Answer 13:We have revised the conclusions.
Question 14:Pls revise the format of the references.
Answer 14:We have revised the format of the references.
Question 15:Line 283, pls revise the order of figures number (Figures 2 and 4)
Answer 15:We have changed the order of the numbering.
We would love to thank you for allowing us to resubmit a revised copy of the manuscript and we highly appreciate your time and consideration.
Hong Zhang
- Liang M, Xin X, Fan W, et al. Phase Behavior and Hot Storage Characteristics of Asphalt Modified with Various Polyethylenes: Experimental and Numerical Characterizations[J]. Construction and Building Materials, 2019, 203(10):608-620.
- Liang M, Ren S S, Fan W Y, et al. Rheological property and sta-bility of polymer modified asphalt: Effect of various vinyl-acetate structures in EVA copolymers[J]. Construction and Building Materials, 2017,137(15): 367-380.
- Yan K Z, Zhe H, You L Y, et al. Influence of ethylene-vinyl acetate on performance improvements of the low-density polyethylene-modified bitumen[J]. J Clean Prod, 2021, 278: 123865.
- Ho, R. Church, K. Klassen, et al. Study of recycled polyethylene materials as asphalt modifiers[J], Canadian Journal of Civil Engineering, 2006, 33(8): 968–981.
- Yan K Z, Liu H, Miljkovi´c M, et al. Influence of ethylene-vinyl acetate on the performance improvements of low-density polyethylene-modified bitumen[J], Journal of Cleaner Production, 2021, 278: 123865.

Reviewer 2 Report
The paper presents interesting research on binders modified with the addition of waste polyethylene and ethylene vinyl acetate (EVA). The aim of the research was to investigate the effect of furforal extract oil, Dcp, KH-570 and CaCO3 on the rheological properties and storage stability of the modified bitumen.
Overall, I think the paper is good.
I have the following comments on the paper:
- statistical analysis of research results should be extended to include the assessment of the significance of variables (factors),
- the analysis of rheological test results should also include a reference to the requirements resulting from the function of the binder in asphalt mixtures and road surface layers,
- this also applies to the conclusions presented in the work,
- the paper lacks the results of research on low temperature performance and it should not be in the abstract.
Author Response
Dear Ms. Irina Mariana and dear reviewer
Re: Manuscript ID: materials-2331236 and Title: Investigation of the rheological properties and storage stability of waste polyethylene/ethylene–vinyl acetate-modified asphalt with Crosslinking and Silicone Coupling Agent
Thank you for your letter and the reviewers’ comments concerning our manuscript entitled “materials-2331236” (ID). Those comments are valuable and very helpful. We have read through comments carefully and have made corrections. Based on the instructions provided in your letter, we uploaded the file of the revised manuscript. Revisions in the text are shown using yellow highlight for additions, and strikethrough font for deletions. We had sought the MDPI appointed company to do the English grammar changes and indicate in purple. The responses to the reviewer's comments are presented following:
Question 1:Statistical analysis of research results should be extended to include the assessment of the significance of variables (factors). The analysis of rheological test results should also include a reference to the requirements resulting from the function of the binder in asphalt mixtures and road surface layers. This also applies to the conclusions presented in the work.
Answer 1:I had modified the conclusion accordingly.
Question 2:The paper lacks the results of research on low temperature performance and it should not be in the abstract.
Answer 2:Our study did not delve into low temperature performance and we have removed the effect of the study on low temperatures from the abstract.
We would love to thank you for allowing us to resubmit a revised copy of the manuscript and we highly appreciate your time and consideration.
Hong Zhang

Reviewer 3 Report
Manuscript, interesting. Rheological studies are very much needed for waste materials.
Well described research scheme.
Post photos of the apparatus and samples during the test.
Provide a legend for designations 1 - 9 in Fig. 6.
Author Response
Dear Ms. Irina Mariana and dear reviewer
Re: Manuscript ID: materials-2331236 and Title: Investigation of the rheological properties and storage stability of waste polyethylene/ethylene–vinyl acetate-modified asphalt with Crosslinking and Silicone Coupling Agent
Thank you for your letter and the reviewers’ comments concerning our manuscript entitled “materials-2331236” (ID). Those comments are valuable and very helpful. We have read through comments carefully and have made corrections. Based on the instructions provided in your letter, we uploaded the file of the revised manuscript. Revisions in the text are shown using yellow highlight for additions, and strikethrough font for deletions. We had sought the MDPI appointed company to do the English grammar changes and indicate in purple. The responses to the reviewer's comments are presented following:
Question 1:Post photos of the apparatus and samples during the test.
Answer 1:For personal reasons I am unable to reach the laboratory at the moment. If you need me, I will return to the laboratory as soon as possible and provide you with the equipment and sample diagrams.
Question 2:Provide a legend for designations 1 - 9 in Fig. 6.
Answer 2:For personal reasons I am unable to reach the laboratory at the moment. If you need me, I will return to the laboratory as soon as possible and provide you with the equipment and sample diagrams.
We would love to thank you for allowing us to resubmit a revised copy of the manuscript and we highly appreciate your time and consideration.
Hong Zhang

Round 2
Reviewer 1 Report
No further comments as all my comments were addressed by the authors